# Thiamine and Biotin: Relevance in the Production of Volatile and Non-Volatile Compounds during *Saccharomyces cerevisiae* Alcoholic Fermentation in Synthetic Grape Must

**DOI:** 10.3390/foods12050972

**Published:** 2023-02-24

**Authors:** Marie Sarah Evers, Chloé Roullier-Gall, Christophe Morge, Celine Sparrow, Antoine Gobert, Stefania Vichi, Hervé Alexandre

**Affiliations:** 1Institut Universitaire de la Vigne et du Vin Jules Guyot, UMR PAM, Université de Bourgogne, 2 Rue Claude Ladrey, 21000 Dijon, France; 2Sofralab SAS, 79 Avenue Alfred Anatole Thévenet, 51530 Magenta, France; 3Food Science and Gastronomy Department, University of Barcelona, Nutrition, INSA (Institut de Recerca en Nutricio I Seguretat Alimentaria), 08921 Santa Coloma de Gramenet, Spain

**Keywords:** thiamine, biotin, vitamins, *Saccharomyces cerevisiae*, alcoholic fermentation, wine, volatiles, metabolomics

## Abstract

Vitamins are major cofactors to numerous key metabolic pathways in enological yeasts, and both thiamine and biotin, notably, are believed to be essential to yeast fermentation and growth, respectively. In order to further assess and clarify their role in winemaking, and in the resulting wine, alcoholic fermentations of a commercial *Saccharomyces cerevisiae* active dried yeast were conducted in synthetic media containing various concentrations of both vitamins. Growth and fermentation kinetics were monitored and proved the essential character of biotin in yeast growth, and of thiamine in fermentation. The synthetic wine volatile compounds were quantified, and notable influences of both vitamins appeared, through a striking positive effect of thiamine on the production of higher alcohols, and of biotin on fatty acids. Beyond the evidence of this influence on fermentations and on the production of volatiles, this work proves, for the first time, the impact held by vitamins on wine yeasts’ exometabolome, investigated through an untargeted metabolomic analysis. This highlighted chemical differences in the composition of synthetic wines through a notably marked influence of thiamine on 46 named *S. cerevisiae* metabolic pathways, and especially in amino acid-associated metabolic pathways. This provides, overall, the first evidence of the impact held by both vitamins on the wine.

## 1. Introduction

Wine is a highly complex and versatile drink, presenting a large variety of chemical compounds that derive from numerous sources during vinification [1] and play significant roles in the sensory profiles and perception of the final product. As such, over 8000 volatile compounds have been estimated to be found in wines, a significant number of those resulting from fermentation processes [2,3]. It has been shown that those generated by yeast metabolism during the alcoholic fermentation are affected by the type of fermenting yeast, as well as the fermentative conditions [4,5,6,7]. As such, nutrient availability in the yeast cultivation medium stands as a significant factor regarding the production of volatile compounds during fermentation [8,9,10,11], susceptible of highly modifying the final aromatic profile in the resulting wine.

Vitamins, consequently, being involved in numerous yeast metabolic pathways, such as those of fatty acids, amino acids, or carbohydrates’ metabolisms, as well as those of sulfur and nitrogen [12], stand as highly significant nutritional compounds. Thus, detrimental effects resulting from their deficiencies in the yeast have been reported, such as impairments of the growth and fermentation kinetics [13,14,15,16] and the production or accumulation of possibly detrimental metabolites [17,18,19,20,21]. Amongst all water-soluble vitamins, biotin and thiamin have been distinguished as particularly significant regarding yeast metabolism: biotin, is, indeed, essential to yeast growth [13,22], while thiamine has been recognized as essential to fermentation processes [15].

However, such investigations remain ancient, and there is no updated knowledge regarding an overall significance of the vitaminic nutrition of yeasts on the production of volatiles during alcoholic fermentation. The aim, here, is therefore to investigate the overall impact of both biotin and thiamin on the course of alcoholic fermentation, assessing their significance regarding kinetics and the production of volatile compounds in a synthetic grape must matrix.

## 2. Materials and Methods

### 2.1. Chemicals, Reagents, and Materials

Cultivation medium: Glucose, fructose, malic acid, citric acid, potassium dihydrogen phosphate, potassium sulfate, magnesium sulfate heptahydrate, sodium chloride, ammonium chloride, tyrosine, tryptophane, isoleucine, aspartic acid, glutamic acid, arginine, leucine, threonine, glycine, glutamine, alanine, valine, methionine, phenylalanine, serine, histidine, lysine, cysteine, proline, manganese (II) sulfate monohydrate, copper (II) sulfate, cobalt (II) chloride hexahydrate, boric acid, ammonium molybdate, sodium hydroxide, myo-inositol, pantothenic acid hemi-calcium salt, thiamine hydrochloride, nicotinic acid, pyridoxine, biotin, ergosterol, oleic acid, and Tween 80 were purchased from Sigma (Merck, Darmstadt, Germany). Calcium chloride dihydrate and sodium bicarbonate were purchased from Honeywell (Charlotte, NC, USA). Zinc sulfate heptahydrate was purchased from Prolabo (VWR, Avantor, Radnor, PA, USA). Potassium iodide was purchased from Merck (Darmstadt, Germany).

Flow cytometry: Phosphate buffer saline (PBS) was purchased from Sigma (Merck, Darmstadt, Germany). The 5-6-carboxyfluorescein diacetate (cFDA) was purchased from Molecular Probes (Thermo Fisher Scientific, Waltham, MA, USA).

Headspace solid-phase microextraction coupled to gas chromatography (HS-SPME-GC/MS): Ethanol 96%, 1-propanol, 3-methylbutyl acetate, 3-methylbutanol, ethyl octanoate, ethyl decanoate, 1-hexanol, 2-phenylethyl alcohol, and octanoic acid were purchased from Sigma-Aldrich (St Louis, MO, USA). SPME fiber divinylbenzene/carboxen/polydimethylsiloxane 50/30 µm, 1 cm-long (DVB/CAR/PDMS), were from Supelco (Bellefonte, PA, USA).

### 2.2. Cultivation Medium

A synthetic must MS300 was used for the fermentations, adapted from Bely and colleagues [23]. The base medium, adjusted at pH 3.3, thus contained the following components (expressed per liter): glucose 100 g, fructose 100 g, DL malic acid 6 g, citric acid 6 g; mineral salts: KH_2_PO_4_ 750 mg, KH_2_SO_4_ 500 mg, MgSO_4_·7H_2_O 250 mg, CaCl_2_·2H_2_O 155 mg, NaCl 200 mg; nitrogen compounds: NH_4_Cl 460 mg, L-tyrosine 18 mg, L-tryptophane 179 mg, L-isoleucine 33 mg, L-aspartic acid 45 mg, L-glutamic acid 120 mg, L-arginine 374 mg, L-leucine 48 mg, L-threonine 76 mg, L-glycine 18 mg, L-glutamine 505 mg, L-alanine 145 mg, L-valine 45 mg, L-methionine 31 mg, L-phenylalanine 38 mg, L-serine 79 mg, L-histidine 33 mg, L-lysine 17 mg, L-cysteine 13 mg; trace elements: MnSO_4_·H_2_O 4 mg, ZnSO_4_·7H_2_O 4 mg, CuSO_4_·5H_2_O 1 mg, KI 1 mg, CoCl_2_·6H_2_O 0.4 mg, H_3_BO_3_ 1 mg, (NH_4_)_6_Mo_7_O_24_ 1 mg; anaerobic growth factors: ergosterol 3 mg, oleic acid 1 mg, Tween 80 100 mg; vitamins: myo-inositol 20 mg, pantothenic acid hemi-calcium salt 1.5 mg, nicotinic acid 2 mg, pyridoxine 0.25 mg.

In order to evaluate the impact of both thiamine and biotin on the production of volatile compounds, this base medium was declined under nine different variations, relying on three different concentration modalities for each of the vitamins: a total absence of the given vitamin, a concentration fit to trigger deficiencies in yeasts [24,25], and the original concentration of the MS300 as described by Bely et al. [23]. The nine different cultivation medium variations are described in Table 1.

### 2.3. Yeast and Inoculation Procedure

A commercial strain of *Saccharomyces cerevisiae* “Selectys^®^ La Marquise” (Sofralab, Magenta, France) was selected for fermentation during this experiment. The active dried yeast (ADY) was rehydrated according to the manufacturer’s specifications, in 10 mL of mineral water, and left to rest for half an hour at 28 °C. The density of viable cells was then determined by flow cytometry on a BD Accuri™ C6 Plus (Becton, Dickinson and Company, Franklin Lakes, NJ, USA). The fluorophore used to detect the viable cells was 5-6-carboxyfluorescein diacetate (cFDA) dissolved in acetone at a final concentration of 1500 µM. Then, 3 µL of this solution was added to 100 µL of the yeast suspension diluted in PBS for the measurement. This yeast suspension was then taken and used to inoculate 200 mL of the selected synthetic must medium at an initial concentration of 10^6^ viable cells/mL.

### 2.4. Fermentation Conditions and Monitoring

All fermentations were carried out in MS300 synthetic must medium, according to the variations described in Section 2.2, previously sterilized by vacuum filtration on a 0.22 µm pored membrane-based apparatus (Millipore Steritop, Merck, Darmstadt, Germany). Erlenmeyer flasks of a 250 mL total volume, containing 200 mL of the sterile cultivation medium and covered with sterile cotton wool to avoid any cross-contaminations, were used for the conduction of the fermentations, and set at 28 °C without shaking. A volume of 500 µL of all samples was collected every 3 h during the first two days, except for the initial first 12 h (lag phase), during which no sampling was performed. Samplings were then collected at 24 h intervals after the completion of the exponential phase. All cultures were conducted in triplicate.

The viable cell concentration was determined similarly to the procedure used for inoculation, thus allowing for the monitoring of biomass growth. Fermentation progress was monitored according to the procedure described by Seguinot and colleagues [9], by weighing of the Erlenmeyer flasks, with the weight loss being assimilated to the amount of CO_2_ released over the course of fermentation. As such, the maximal rate of CO_2_ production, or the maximal fermentation rate, was calculated by the derivation of CO_2_ production over time [9].

### 2.5. Analytical Methods

Endpoint fermentation samples were centrifuged at 13,000 rpm for 3 min, and the collected supernatants were stored at −20 °C until analysis.

#### 2.5.1. Enzymatic Determination of Central Carbon Metabolites

Acetic acid, acetaldehyde, D-lactic acid, and glycerol were enzymatically quantified using a Y15 enzymatic autoanalyzer (Biosystems, Barcelona, Spain), calibrated and configured for the associated enzymatic kits (references 12930, 12820, 12812, 12801; Biosystems, Barcelona, Spain). Ethanol was determined by Fourier Transform Infrared (FTIR) spectroscopy with an OenoFoss type 4101 apparatus (FOSS Electric, Hillerød, Denmark).

#### 2.5.2. HS-SPME-GC/MS Analysis of Volatile Compounds

Extraction of volatiles was carried out by means of a Combi-pal autosampler (CTC Analytics, Zwingen, Switzerland). A volume of 2 mL of sample was placed into a 10 mL vial that was then fitted with a polytetrafluoroethylene (PTFE)/silicone septum and maintained under continuous agitation (250 rpm) at 40 °C during 10 min for sample conditioning. Then, a DVB/CAR/PDMS fiber was exposed to the sample headspace at the same temperature and stirring during 30 min and immediately desorbed in the gas chromatograph injector.

Volatile compounds were analyzed by gas chromatography coupled to quadrupolar mass selective spectrometry using an Agilent 5973 Network detector (Agilent Technologies, Palo Alto, CA, USA). Analytes were separated on a Supelcowax-10 (Supelco), 60 m × 0.25 mm I.D., with a 0.25 μm film thickness. Column temperature was held at 40 °C for 10 min, increased to 150 °C at 3 °C/min, then to 250 °C at 15 °C/min, holding for 5 min. The injector temperature was 260 °C and the time of desorption of the fiber into the injection port was fixed at 5 min. Helium was the carrier gas, at a flow rate of 1.5 mL/min. The temperature of the ion source was 230 °C and of the transfer line was 280 °C. Electron impact mass spectra were recorded at 70 eV ionization energy, 5.1 scan/s. GC–MS analysis was performed in the complete scanning mode (SCAN), in the 35–300 u mass range.

Identification of compounds was carried out by comparison of their mass spectra and retention times with those of standard compounds or with those available in the mass spectrum library Wiley 6 and in the literature, respectively. Response factors of volatile compounds were calculated using a calibration curve, obtained by analyzing a hydroalcoholic solution (ethanol 10%, *v*/*v*) with different concentrations of reference compounds.

#### 2.5.3. UHPLC-Q-ToF-MS Non-Targeted Analysis

UHPLC-Q-ToF-MS non-targeted analyses of the yeast metabolome were performed using an ultra-high-pressure liquid chromatography (UHPLC) Dionex Ultimate 3000 apparatus (Thermo Fisher Scientific, Waltham, MA, USA) coupled to a MaXis plus MQ ESI-Q-ToF mass spectrometer (MS) (Bruker, Bremen, Germany).

Non-polar compounds were eluted in reverse phase on an Acquity BEH C18 1.7% m, 100 × 2.1 mm column (Waters, Guyancourt, France), using mobile phases 0.1% MS grade formic acid in ultrapure water (Milli-Q, Merck, Darmstadt, Germany) (eluent A) and 0.1% formic acid in 95% MS grade acetonitrile (Biosolve, Dieuze, France) (eluent B). The elution was performed as a gradient at a flow rate of 400 µL/min at 40 °C, with its initial 5% eluent B phase being held for 1.10 min, before increasing to reach 95% eluent B at 6.40 min until the end of the elution after a total 10 min analytical run.

The nebulizer pressure was set at 2 bar, the nitrogen dry gas flowing through the apparatus at 10 L/min, while ionization was performed as electrospray in either positive or negative mode. Ion transfer parameters were set at 500 V as endplate offset, and 3500 or 4500 V as capillary voltage for negative and positive ionization mode, respectively. The mass spectrometer system was calibrated prior to analyses using sodium formate in enhanced quadratic mode (errors < 0.5 ppm). Mass ranges stood between 100 and 1000 *m*/*z*, while the stability of the UHPLC-Q-TOF-MS-MS system was ensured through the analysis of quality controls, prepared as a mix of all the analyzed samples, at the beginning, the end, and every ten samples during the analysis. All the samples were randomly analyzed, and the injection of the sodium formate calibrant at the beginning of each run allowed recalibrations of the spectrum.

Software Bruker Compass MetaboScape (v. 8.0.1, Bruker, Mannheim, Germany) was used for pre-treatment analysis comprising: the internal mass recalibration of the spectrum, extraction of molecular features (*m*/*z* couple, retention time), and alignment and annotation using SmartFormula (isotopic profile). Feature extraction has been realized for an absolute intensity higher than 1000 and the presence of the feature in more than 20% of the total number of samples. Recursive extraction has been used to find features without an intensity threshold if it presents in more than 20% of samples. SmartFormula annotation has been realized with the criteria: mass error < 10 ppm and mSigma < 20.

Isolated significant features were assigned possible annotations using the MassTRIX application [26], and its associated databases, such as KEGG [27], as accessed on 11 November 2022.

### 2.6. Statistical Analyses

Data analyses were performed with a statistical treatment and graphically (PCA biplots, chromatograms, boxplots) represented using the R software (version 4.0.3) (R foundation, Vienna, Austria). Nonparametric Scheirer–Ray–Hare (chosen as a two-way ANOVA equivalent) and Dunn (relying on a Bonferroni correction) tests from the rcompanion [28] and rstatix package [29] were used for statistical comparison of the growth and fermentation kinetics data, as well as of the samples resulting from the quantification of volatiles and carbon metabolites in wines. A *p*-value < 0.05 was used as the acceptance criterion for rejecting the null hypothesis. PCA and one-way ANOVA were used for statistical analysis of the metabolomics datasets, at confidence levels of α = 5% and α = 1%, respectively. Heatmaps in the volatilome and exometabolome investigations were drawn for biotin and thiamine on all their respective significantly affected compounds or features.

## 3. Results and Discussion

### 3.1. Impact on the Growth and Fermentation Kinetics

Both thiamine and biotin were evaluated for their ability to support the growth of *S. cerevisiae* in MS300 synthetic must. Significant differences in growth kinetics were observed due to the initial concentration of biotin present in the cultivation medium. Three growth parameters were assessed during the experiment: the maximal specific growth rate (µ_max_) attained during the exponential phase, the generation time (G), the maximal population reached during cultivation (y_max_), the duration of the alcoholic fermentation process (t_AF_), and the maximal production rate of CO_2_ during fermentation (r_CO2max_). The ability of yeasts to complete the fermentation processes, with complete consumption of sugars in the medium, was also evaluated.

A strong dose-related effect of biotin on growth appeared throughout yeast monitoring, with significant increases (*p* < 0.05) arising for all growth parameters (Table 2, Figure 1, Appendix A). Notable rises in the µ_max_ and y_max_ values, as well as a significant drop in the generation time G values, can be found for initial biotin concentrations as little as 0.5 µg/L in the growth medium, suggesting that low doses of biotin might be sufficient in ensuring the coverage of the yeast requirements in the vitamin.

Such an influence stands as both in regard to the growth celerity, as well as its intensity, allowing for the achievement of higher yeast populations when biotin is sufficiently provided in the growth medium. A saturation effect appeared here, since no further significant increase (*p* > 0.05) in the yeast maximal population can be found between cells grown in 0.5 µg/L or 3 µg/L of biotin (Table 2, Figure 1, Appendix A). However, no significant effect of biotin on the fermentation rates could be found (Table 2, Figure 1), suggesting that the vitamin solely affects growth, and that its deficiencies do not impair the fermentative capacity of the yeast, although in higher populations. While concurrent conclusions of this matter were drawn on anaerobically grown *S. cerevisiae* yeasts in previous investigations [30], other works rather concluded in, opposingly, a significant influence held by the vitamin on fermentation rates [13,31]. Although the yeasts used in the assays led by Bohlscheid and colleagues were pre-starved in a biotin-deficient medium before inoculation, which might have contributed to inducing the notable differences in behaviors observed here [31], those used by Ough and Kunkee were rehydrated active dried yeasts, in a similar fashion to the yeasts used in the present work [13]. The essential character of biotin appears to be, however, a strain-dependent character in *S. cerevisiae*, since all strains do not possess the capacity for biotin de novo biosynthesis [32]. It is not impossible, as such, that the difference in fermentative behaviors that appears here might rather reflect such a disparity in regard to biotin biosynthesis capacity. There is, however, an overall consensus of past and present findings regarding the essential role assumed by biotin in yeast growth and viability, its significance being, as such, asserted.

In contrast to biotin, thiamine appeared to exert an effect solely on fermentation kinetics and had no impact on the considered growth parameters (Table 3, Figure 1, Appendix A). Interestingly, the significance of thiamine in regard to fermentation does not appear to be dose-related in a linear fashion, since distinctive differences in the fermentation rates only appeared between the yeasts cultivated in 50 µg/L and 250 µg/L of thiamine, the total absence of the vitamin rather standing as an intermediate, and highly versatile condition. While rising concentrations of the initial thiamine in the medium allowed for dose-dependent decreases in the duration of the fermentation process, this dose-dependent trend did not appear in the fermentation rates, since yeasts fermented in 50 µg/L appeared to display the lowest fermentation rates at any given biotin level.

The intermediate fermentation rates resulting from the complete absence of thiamine in the cultivation medium might result from the establishment, by the yeast, of a survival strategy to resist the inadequate growth conditions. It has indeed been shown that, when grown in the complete absence of thiamine, *S. cerevisiae* displays thiamine synthesis-related proteins Thi4 and Thi5 as amongst its most abundant proteins [33], both acting as suicide enzymes to maintain thiamine levels [34]. Interestingly, *THI4* expression has been positively correlated to the *S. cerevisiae* fermentation rate [35], as well as negatively correlated to the thiamine concentration in the medium [36]. Overall, the overexpression of *THI4* has been proven to improve glucose fermentation and the resulting ethanol production [37]; as such, it could be envisioned that, in the absence of thiamine, a metabolic balancing system is established within the yeast as a response to the deficiency, allowing it to maintain the sufficient fermentative capacity that has been observed here.

Surprisingly, however, these findings regarding the role exerted by thiamine on both *S. cerevisiae* growth kinetics do not agree with previous conclusions found on the matter by Labuschagne and colleagues; as such, notable decreases in *S. cerevisiae* EC1118 growth rates have been noted in thiamine limitation conditions, with significant effects observed for doses lower than 125 µg/L [38], while thiamine, whether absent or as low as 50 µg/L, did not induce any growth impairment here. Although these findings regarding the impact held by thiamine on the yeast growth do not concur, the research conducted by Labuschagne and colleagues does agree with the present results, that conclude in the essential character of thiamine for alcoholic fermentation [38]. A difference in fermentative behavior in regard to thiamine does, however, appear since increases in thiamine have led to rather proportional increases in the yeast fermentation capacity in the work conducted by Labuschagne [38], rather than inducing a maintained fermentation state in complete absence of thiamine, as observed here. Such diverging findings might be an expression of individual differences between the different strains used, or might reflect the differences in physiological states of the yeasts at inoculation; as such, the fermentations conducted by Labuschagne and colleagues relied on thiamine-starved, pre-cultivated yeasts, rather than directly rehydrated commercial ADY that were used in the present experiment, and specifically chosen to replicate common yeasting conditions that may be found in winemaking contexts. Such differences in the yeasts’ physiological states might have impacted their behavior in regard to thiamine and led to the observed differences. Plus, the constant agitation applied on the fermentation broth during the assays led by Labuschagne and colleagues, opposingly to the static mode selected here, might have notably impacted thiamine assimilation by the fermenting yeasts, as it does other nutrients [39,40].

All in all, those results highlight the essential character held by biotin on yeast growth, allowing for it to reach sufficient population and rates for proper course of the biological processes. Similarly, thiamine’s significance in regard to fermentation [15] has been further assessed here, although a complete thiamine deprivation has been associated with maintained fermentation rates, possibly through the establishment of a survival strategy by the yeast, or the possibility that *S. cerevisiae* might have had some leftover inner thiamine resources that have allowed it to cope with thiamine deprivation.

### 3.2. Impact on Central Carbon Metabolites

In order to further assess the impact held by both vitamins on the yeast metabolism, acetaldehyde, acetic acid, D-lactic acid, and glycerol were enzymatically quantified on the endpoint fermentation supernatants for all wines, and ethanol was determined through FT-IR spectroscopy (Table 4 and Table 5).

Consistent with the evidence that all fermentations managed to achieve a complete consumption of sugars, ethanol did not display any significant dependence (*p* > 0.05) in regard to the initial concentrations of the vitamins (Table 4 and Table 5).

Interestingly, amongst all four investigated compounds, solely acetaldehyde and ethanol did not display any significant variation (*p* > 0.05) regarding the initial thiamine concentration in the medium (Table 5). Such an effect on acetaldehyde does contradict previous knowledge, that established thiamine deficiencies to be associated with higher contents of sulfur-binding compounds, including acetaldehyde [41], as well as the strong dependency manifested by pyruvate decarboxylase (Pdc) towards thiamine [42] for the formation of acetaldehyde from pyruvate. However, some other findings, although recognizing the clear influence of thiamine deficiencies on sulfur-binding ketoacids, did not, opposingly, find any impact of it on acetaldehyde itself [43,44]. Here, the absence of any effect of thiamine on acetaldehyde contents in wines clearly appeared. Biotin, however, appeared to have a significant effect on acetaldehyde accumulation (*p* < 0.05) (Table 4). Such a rise in acetaldehyde contents might indirectly result from other influences held by the vitamin on carbon metabolism, although its exact cause remains obscure.

In a surprising fashion, acetic acid displayed significant (*p* < 0.05) and opposed responses to both vitamins (Table 4 and Table 5); as such, while it appears to increase concurrently with the initial thiamine concentration in the medium, its contents seem to drop when increasing the initial biotin. Interestingly, however, the statistical interaction between both vitamins, by itself, does not hold any significant influence (*p* > 0.05) on the acetic acid contents formed during alcoholic fermentation. The positive effect held by the initial thiamine on acetic acid here appears consistent with previous conclusions, that found higher concentrations of acetic acid to be associated with higher thiamine [20]. Although the negative effect held by the initial biotin on acetic acid contents appears to be more surprising, evidence show that there is a lesser incorporation of acetic acid into fatty acids in biotin-deficient yeast cells [45], suggesting that, in the absence of the vitamin, there might be an accumulation of acetic acid in the medium, leading to the higher contents that were observed here.

While biotin did not have any significant influence (*p* > 0.05) on the D-lactic acid contents of the produced wines, a clear effect of thiamine (*p* < 0.05) on those contents appeared (Table 4 and Table 5). Interestingly, D-lactic acid concentrations seemed to be decreasing against the initial thiamine contents. This negative effect most likely results from increases in lactic acid production favored by thiamine depletions due to the defective thiamine pyrophosphate (TPP)-dependent pyruvate decarboxylase (Pdc) activity [46], that therefore redirects pyruvate towards lactic acid synthesis.

Finally, glycerol contents in the final wines displayed a significant dependency (*p* < 0.05) towards both the initial biotin and thiamine concentrations in the growth medium (Table 4 and Table 5); as such, the amounts of glycerol produced noticeably increased when increasing both vitamins. This appears consistent with previous findings, that concluded towards drops in the glycerol contents when fermenting in thiamine-deficient media [47], while the effect of biotin on glycerol production has not been assessed yet. The statistical interaction between both vitamins, however, did not display any significant influence (*p* > 0.05) on glycerol production.

All in all, the consequent involvement of thiamine in the yeast central carbon metabolism (CCM) is potently highlighted here, as a reflection of its cofactor role towards several CCM-associated enzymes [42]. While the effect held by biotin on these metabolites appeared slightly less consequent, its noticeable influence on both acetaldehyde and acetic acid emerged here, overall concluding towards the clear existence of an effect of both vitamins on the yeast metabolism during alcoholic fermentation.

### 3.3. Relevance for the Production of Volatile Compounds during Alcoholic Fermentation

The volatile profiles of wines resulting from the alcoholic fermentation of *S. cerevisiae* on the medium containing different concentrations of both thiamine and biotin were analyzed by HS-SPME-GC/MS, with the aim of assessing any potential effect held by those vitamins on the wine volatilome. Significant differences between the various vitamin doses have been investigated through Scheirer–Ray–Hare nonparametric tests, and hierarchical clustering of the compounds that displayed a significant reliance for either biotin or thiamine was performed using the average concentrations for those compounds at each considered vitamin level (Figure 2). Amongst the 31 compounds that were analyzed, 13 displayed a significant influence (*p* < 0.05) of the initial biotin composition, 11 were significantly correlated (*p* < 0.05) to the initial thiamine concentration, and only 1, methyl heptenone, was significantly impacted by the interaction between both vitamins, but interestingly, not by thiamine or biotin considered individually (Table 5 and Table 6). It is also relevant to note that only isovaleric acid was impacted by both initial thiamine and biotin individually.

#### 3.3.1. Biotin Influence on Wine Volatile Compounds

Out of the 31 analyzed, 13 volatile compounds were significantly influenced by the initial biotin concentration in the synthetic must medium (Table 6, Figure 2A–C), offering the first evidence for the relevance of biotin in the production of wine volatiles, on a substantial scale of analyzed compounds.

##### Fatty Acids, Fatty Alcohols, and Fatty Acid Ethyl Esters (FAEEs)

A clear effect held by the vitamin on fatty acid metabolism appeared here, since 7 of the 13 fatty acids, fatty alcohols, and fatty acid ethyl esters (FAEEs) displayed significant (*p* > 0.05) dose-related variations in their contents (Figure 2A).

Notable increases in all the fatty acid-related compounds appeared when increasing the initial biotin concentration in the medium, therefore highlighting a strong stimulatory effect of the vitamin on those compounds. Interestingly, it must be noted that such increases do appear for all derived compounds of a given fatty acid, with its associated alcohol and ethyl esters being heightened in a similar fashion, suggesting that the stimulatory influence of the vitamin on those compounds appears at the previous stages of the fatty acid synthesis pathway. This effect is to be explained by the essential role played by biotin in fatty acid synthesis and elongation, with the vitamin acting as a cofactor to Acc1 and Hfa1 in the conversion of acetyl-CoA towards malonyl-CoA, which stands as the first step of the fatty acid synthesis pathway, before malonyl-CoA enters the FAS system [48,49,50]. Ethyl nonanoate, however, did not present any significant effect of the initial biotin on its concentration (0.39 ± 0.05 µg/L, 0.48 ± 0.17 µg/L and 0.52 ± 0.16 µg/L at 0, 0.5, and 3 µg/L of biotin, respectively), although its precursor, nonanol, did indeed show similar increases to the other impacted compounds, although to an apparently lesser degree (Figure 2). Such a difference might be solely resulting from the decreased transfer of ethyl esters to the medium when increasing the fatty acid carbon chain length [51]. However, the other longer-chained fatty acids and associated alcohols and FAEEs, such as decanoic acid (C10), decanol, ethyl decanoate, and ethyl laurate (C12), displayed no significant differences in the concentrations in regard to the initial biotin contents of the medium.

Three of the impacted compounds, ethyl hexanoate, ethyl octanoate, and octanoic acid, do, interestingly, display odor activity values (OAV) above 1, detected above their olfactory thresholds of 14 µg/L, 5 µg/L, and 5 mg/L, respectively [52]. This suggests that the initial biotin in the medium might lead to notable changes in the olfactory profiles of wines. No definite conclusion, however, can be drawn on the matter without any sensory analysis to definitely assess the impact held by the vitamin on wine aromatic profiles; this, however, strikingly proves the relevance of sufficient biotin in grape must for the production of fatty acids and FAEEs.

##### Higher Alcohols, Higher Aldehydes, Fusel Acids, and Associated Esters (Ehrlich Pathway)

In addition to the clear impact held by the initial biotin on the production of fatty acids and derivatives that has been observed here, a lesser influence of the vitamin on the production of higher alcohols and derivatives has appeared, since 4 of the 10 higher alcohols, higher aldehydes, and associated esters, that can be linked to the Ehrlich pathway, display significant differences in their contents in regard to the initial biotin contents of the synthetic must (Figure 2B). Interestingly, two notable groups of affected compounds seemed to materialize in regard to their behaviors in the face of the initial biotin concentration: while propyl acetate and isovaleric acid increased in contents when increasing biotin, isoamyl alcohol and isoamyl acetate appeared to decrease against the vitamin (Figure 2B).

Surprisingly, no consensus appears to exist in regard to the higher levels of isoamyl alcohol and its corresponding acetate observed here; as such, albeit Bohlscheid and colleagues reported decreases of the isoamyl alcohol contents in case of biotin deprivation [31], a complete absence of the effect of the vitamin on the production of this compound had been previously reported by Gutierrez and associates [53]. With the yeast strains used in these various studies being different, it is not impossible that this disagreement in behavior could rather be a strain-dependent character, or reflect their individual tolerances towards low biotin availabilities. This could also result from disparities in the yeasts’ physiological states, since, notably, the yeasts used by Bohlscheid and colleagues were pre-starved in a biotin-deficient medium before inoculation [31].

The increase in the other compounds when increasing biotin might rather be related to their amino acid origin. Reductions in the cellular amino acid contents of yeasts have been found when grown in biotin-deficient medium [54], which might justify the limited formation of their associated higher alcohols, aldehydes, and esters in case of biotin deficiency. Although the exact metabolic impact of biotin in the production of these compounds is not clear, such a behavior can be overall associated with the role held by the vitamin as a cofactor to several carboxylases, including pyruvate and acetyl-CoA carboxylases as well as in the Ehrlich pathway decarboxylation step of α-keto acids towards their associated aldehydes [49,50,55].

Amongst these impacted contents, both isovaleric acid and isoamyl alcohol were found in concentrations above their odor detection thresholds of 0.7 and 7 mg/L, respectively [52], further supporting the idea that biotin might play a role in the sensory properties of final wines.

Interestingly, the ratio between the total ethyl esters (calculated as sum of the ethyl esters quantified here) and total acetate esters (calculated as sum of the acetate esters quantified here) appears to be significantly (*p* < 0.05) influenced by the initial biotin in the medium, increasing alongside the vitamin. Surprisingly, though, the total esters (calculated as sum of the esters quantified here) do not present any significant (*p* > 0.05) dependency regarding biotin, suggesting that the vitamin indeed modulates the ester profile of the wines, orienting it preferentially towards ethyl esters rather than acetate esters, and offering the first evidence of such an effect of the vitamin on the ester formation in wine.

##### Central Carbon Metabolism-Derived Compounds

Notable increases in the diethyl succinate concentrations were found when increasing the initial biotin contents in the synthetic must medium (Table 6, Figure 2C), rising from 0.35 ± 0.05 to 0.45 ± 0.03 µg/L when increasing biotin from 0 to 3 µg/L.

On the other hand, significant decreases in the concentration of benzaldehyde formed during fermentation appeared when raising the initial biotin doses (Table 6, Figure 2), dropping from 52.50 ± 16.62 µg/L to 29.62 ± 4.91 µg/L when reaching 3 µg/L of the initial vitamin in the synthetic must.

All in all, such results further suggest the influence that biotin might have on the yeast central carbon metabolism during alcoholic fermentation.

#### 3.3.2. Thiamine Influence on Wine Volatile Compounds

Out of the 31 analyzed, 11 volatile compounds were significantly influenced by the initial thiamine concentration in the synthetic must medium (Table 7, Figure 2D–F).

##### Fatty Acids, Fatty Alcohols, and Fatty Acid Ethyl Esters (FAEEs)

Interestingly, amongst those affected compounds, a noticeable impact of the initial thiamine on fatty acids and derivatives appeared here: 5 of the 13 fatty acids, fatty alcohols, and FAEEs indeed displayed significant (*p* < 0.05) dose-related variations in their contents (Table 7, Figure 2D). Surprisingly, only the longer medium-chain fatty acids (MCFAs) that were analyzed here exhibited a significant dependence towards the initial thiamine in the medium, since solely decanoic acid (C10), decanol, ethyl decanoate, and ethyl laurate (C12) were affected here. Those compounds were associated with increases alongside the initial thiamine concentration in the medium (Figure 2D), and the results appear consistent with previous findings by Labuschagne during the alcoholic fermentation of *S. cerevisiae* [38]. On the other hand, butanol and ethoxy propanol were associated with notable decreases in concentrations when increasing the initial thiamine, in accordance with previous findings by Labuschagne [38].

Interestingly, out of those five significantly impacted compounds, solely ethoxy propanol was present in concentrations above its 100 µg/L detection threshold [52], suggesting that the initial thiamine might contribute to modulating the sensory profiles of wines.

##### Higher Alcohols, Higher Aldehydes, Fusel Acids, and Associated Esters (Ehrlich Pathway)

Besides the influence held by the vitamin on fatty acid metabolism, an impact of the initial thiamine concentration in the synthetic must medium on higher alcohols, higher aldehydes, and their derivative esters appeared here (Table 7, Figure 2E), since 5 of those compounds out of their total 10 displayed significant (*p* < 0.05) variations in regard to thiamine contents. As such, a clear increase in the final concentrations of phenyl ethyl alcohol, phenyl ethyl acetate, and isovaleric acid appeared when raising thiamine doses (Figure 2E). Such a phenomenon appears consistent with the thiamine-dependency exhibited by the Tkl1/2 transketolase of the pentose phosphate pathway (PPP), as well as the decarboxylation step (Aro10, Pdc1/2/5) of the Ehrlich pathway [42]. The Ehrlich pathway precursor phenylpyruvate is, indeed, a product of the shikimate pathway, itself deriving from the PPP through phosphoenolpyruvate, and further converted to phenyl ethanol. It is, however, most surprising that the other higher alcohols do not display a similar behavior and dependency in regard to thiamine, since isoamyl alcohol and isoamyl acetate did not express any significant effect of the initial vitamin on their contents (Figure 2E), consistent with the absence of any influence of the initial thiamine on isoamyl alcohol, as found by Labuschagne using *S. cerevisiae* [38]. Amongst those compounds, both isovaleric acid and phenyl ethanol were found in concentrations above their olfactory detection thresholds of, respectively, 7 and 14 mg/L [52].

It also appears relevant to note that the ratio between superior alcohols and the total esters (calculated as the sum of the individual esters quantified here) was significantly (*p* < 0.05) impacted by the initial thiamine in the medium, and diminished against thiamine doses, suggesting that, although the vitamin stands as a cofactor in the initial steps of the Ehrlich pathway, it overall favors the production of esters in the final wine. Significant (*p* < 0.05) increases in the total esters were indeed found in regard to the initial thiamine, rising from 42.5 ± 0.9 to 71.6 ± 1.8 mg/L when increasing thiamine from 0 to 250 µg/L. This first evidence of an influence held by the vitamin on ester production in wines suggests that sufficient thiamine doses, therefore, might contribute to expanding the ester profiles of wines, and, as such, possibly take part in the definition of the aromatic signature of the wine, since most esters are associated with fruity, pleasant notes [41]. Although the individual esters reported here were mostly under their detection thresholds, and therefore individually might not play any role in the sensory profiles of wines, their synergistic, combined effect might prove significant [41].

##### Central Carbon Metabolism-Derived Compounds

In addition to what was observed in volatiles from the fatty acid metabolism and the Ehrlich pathway, thiamine appeared to have a significant influence on only one carbon metabolism-derived compound out of the four quantified here (Figure 2F).

As such, ethyl benzaldehyde rose from 2.89 ± 0.35 µg/L to 3.49 ± 0.33 µg/L when similarly increasing thiamine from 0 to 250 µg/L, which might be originating from the dependency held by benzaldehyde lyase (BAL) towards thiamine pyrophosphate for the synthesis of acetoin from benzaldehyde and acetaldehyde [56], although benzaldehyde itself, interestingly, did not present any significant differences in regard to initial thiamine. This might, however, be rather dependent of acetaldehyde contents, since those appear to have decreased in an opposite fashion to ethyl benzaldehyde. It could be envisioned, as such, that since the conversion of benzaldehyde and acetaldehyde towards acetoin cannot be as efficiently performed in case of thiamine limitation, the esterification of benzaldehyde towards ethyl benzaldehyde could preferentially occur.

### 3.4. Biotin and Thiamine Impact on the Wine Metabolome

An untargeted UPLC-qTOF-MS analysis of the wines collected at the endpoint of the alcoholic fermentation was performed in order to investigate the influence held by both biotin and thiamine on the *S. cerevisiae* metabolome.

The retrieved features were filtered to only retain those existing in at least two of the three biological replicates of all modalities, to ensure significance in their presence in the final wines. Subsequently to those treatments, 3870 features have been detected in both positive and negative ionization modes and analyzed using both PCA at a significance level of α = 0.05 and one-way ANOVA at a significance level of α = 0.01.

In order to investigate possible differences between the metabolomes of *S. cerevisiae* when grown in different initial concentrations of either biotin or thiamine, a principal component analysis (PCA) was performed on the intensities obtained for all 3870 extracted features, subsequently leading to planar representations in which components PC1, PC2, and PC3 accounted for, respectively, 20.4%, 13.5%, and 5.6% of the observed variations (Figure 3A,D).

No clear discrimination between groups appeared here, and especially no differentiation could be found between the intermediary and lowest vitamin doses. As such, for all further metabolomic investigations, the wines obtained from *S. cerevisiae* grown in 3 µg/L of biotin (“high biotin”) were separated from those grown in both 0.5 and 0 µg/L of biotin (“low biotin”), for which data were taken into account together, as one unique condition. Similarly, wine resulting from alcoholic fermentation in 250 µg/L of thiamine (“high thiamine”) was considered separately from those obtained in both 50 and 0 µg/L of thiamine (“low thiamine”).

#### 3.4.1. Biotin Influence on the Wine Metabolome

ANOVA performed on the extracted features allowed for the isolation of 208 compounds that presented significant differences (*p* < 0.01) between both high and low biotin conditions, therefore accounting for less than 10% of the extracted features. Putative annotations were assigned to all significant extracted features according to the KEGG, Metlin, and Oligonet online databases and tools [27]. Consequently, out of those 208 notable features, 164 were assigned a possible chemical formula, and therefore annotated at level 4 [57]; as such, those annotated features were the sole ones to be considered in all further investigations of the impact of biotin on the *S. cerevisiae* metabolome.

Amongst those, a remarkably higher proportion of features appeared to be significantly more intense in high biotin than in low biotin, since 133 of those appeared significant (*p* < 0.01) in wines obtained from 3 µg/L of biotin, against 31 resulting from those associated with the lower doses of the vitamin. Hierarchical clustering of those features led to a clear distinction between wines obtained from higher and lower initial contents of the vitamin, although two low biotin samples appeared less properly discriminated from the high biotin ones (Figure 3B), which strongly highlights the impact of biotin on the yeast metabolism.

To further assess the nature of this influence, Van Krevelen diagrams were plotted on the features’ O/C and H/C ratios for both biotin conditions, and the elemental compositions of them were investigated (Figure 4). Notable differences in the chemical composition of wines appeared between the features associated with each biotin condition; as such, while high biotin-resulting wines displayed a dominance of CHON-based features, the predominating ones in low biotin wines were the phosphorus-containing features (CHOP, CHONP, and CHONPS). It is, however, relevant to note that those predominant features are, in either biotin condition, found in the lipid (O/C < 0.6 and H/C > 1.3) and polyphenol (O/C < 1.2 and H/C < 1.3) regions of the Van Krevelen diagram, consistent with their predicted chemical families [58]. In addition, an interesting influence exerted by biotin on the proportion in the CHONS features appeared here, since they dropped from 11.3% in high biotin to 6.5% in low biotin.

Subsequently to the features’ nature assumption, hypothetical annotations were assigned using the MassTRIX application [26] (Helmholtz Zentrum München, München, Germany), and its associated databases, such as KEGG [27], as accessed on 11 November 2022.

A limited number of the specific biomarkers were successfully annotated in the databases, since less than 10% of those were matched, reflecting the high complexity and current low understanding of the wine composition [59,60]. Those annotations have allowed to identify the metabolic pathways associated with the changes in the exometabolome changes observed as a result of the initial biotin in the medium (Appendix A). Unsurprisingly, a higher number of pathways were impacted by a high initial biotin, although the number of associated biomarkers to each of those pathways remains limited, as a consequence of the low number of annotated features overall. No pathway appeared here, as such, to be notably more affected than others; however, it appears relevant to note that the impacted metabolic pathways here are reliant on major pathways, such as central carbon metabolism, amino acid metabolism, and lipid metabolism. While its influence on the CCM might be a reflection of the biotin-dependent enzymes Acc1/Hfa1 and Pyc1/2 [42], its influence on amino acids appears less clear, although they might be an indirect result of these biotin-dependent reactions, and notably the Pyc1/2 conversion of pyruvate towards oxaloacetate [42]. The absence of any more significance in the effect of high biotin on the lipid metabolism appeared, however, highly surprising, notably in regard to the essential role held by the vitamin in the first steps of the synthesis of fatty acids [48,49,50]. A similar impact on carbon metabolism was found in the low biotin wines, through the fructose and mannose metabolism. Surprisingly, low biotin was also found to impact riboflavin metabolism.

All in all, the untargeted approach allows to conclude, for the first time, on the actual influence of the initial biotin on the yeast exometabolome during alcoholic fermentation.

#### 3.4.2. Impact of Thiamine on the Wine Metabolome

In order to determine the extent of the influence of thiamine on the yeast exometabolome, an ANOVA was performed on the 3870 extracted features, and resulted in the isolation of 515 that were associated with significant differences (*p* < 0.01) in regard to the initial concentration of the vitamin, therefore accounting for nearly 15% of the extracted features. However, only 378 of those specific features were assigned possible chemical formulas through putative annotations in online databases, in a similar fashion to what was performed on biotin-associated features, and these were the sole ones to be considered for their significant influence in the *S. cerevisiae* thiamine-dependent metabolome. It is also relevant to note that the number of features associated with thiamine here was significantly higher than those associated with biotin, since it more than doubled its amount, also suggesting a greater influence of thiamine on the yeast metabolism.

Similar to what was observed in the assessment of the influence of biotin on the yeast exometabolome, a high proportion of those features were significantly more intense in high thiamine, amounting to 378, while only 69 features appeared significantly more intense (*p* < 0.01) in low thiamine wines. Such a differentiation also appeared quite clearly through hierarchical clustering of these 515 total biomarkers, since it resulted in a distinct discrimination between high and low thiamine wines (Figure 3D), strongly testifying on the influence exerted by the vitamin on the wine exometabolome.

An investigation of the chemical composition of the thiamine-affected features was led, in order to find the precise nature of the effect exercised by the vitamin on the yeast metabolism during fermentation. As such, Van Krevelen diagrams were drawn based on the O/C and H/C atomic ratios of the specific features for both the low and high thiamine conditions, and their elemental composition was investigated in an effort to define their chemical identities. Surprisingly, as opposed to what was observed for biotin, there was no clear difference between the elemental composition proportions of the high and low thiamine features (Figure 5), although there was a slight increase in the percentage of CHO markers in high biotin, amounting to 11.7% of the features, against 7.2% in low thiamine, as well as a slight increase in phosphorus-containing ones in low thiamine, reaching 39.1% of the features, against 33.0% in high thiamine.

Interestingly, however, these CHO features in high thiamine conditions appeared to be amongst the most intense ones, found mostly where the amino sugars (O/C > 0.6 and H/C > 1.5) and carbohydrates (O/C > 0.8, 1.6 < H/C < 2.7) were expected on the Van Krevelen diagrams. Plus, it appears relevant to note that the strongly represented CHON biomarkers, not significantly varying in proportion between both thiamine conditions, also appeared to display similar behaviors in regard to their intensities and nature, whether obtained from low or high thiamine wines. They are, indeed, associated with a number of high intensities, as observed on Van Krevelen diagrams, and both were found in the area in which polyphenols were expected (O/C < 1.2 and H/C < 1.3), consistent with the high proportion of polyphenol compounds that were predicted [58]. Thiamine did not appear, consequently, to interfere in any way with the frequency of formation of neither polyphenols nor lipids; however, the high thiamine exometabolome did display increases in the proportion of carbohydrates and amino sugars formed during alcoholic fermentation, while low thiamine appeared to induce noticeable increases in the proportion of synthetized proteins.

A limited number of biomarkers for both thiamine conditions was annotated at level 3 [57] in online databases, since less than 5% of those were matched with an identification. However, this further annotation process has led to the identification of the pathways in which those biomarkers are involved, and therefore, associated with the exometabolome changes that were observed here (Appendix A). In contrast to the low number of pathways affected by the initial biotin, there was, here, a thorough influence of high thiamine on numerous metabolic pathways, since 46 named pathways were matched through KEGG [27] for its associated biomarkers, while a far lesser number of metabolic pathways were influenced by low thiamine. Interestingly, high thiamine seems to exert a strong influence on both amino acid and carbon metabolisms, since they displayed the highest number of total annotated biomarkers (53 and 51, respectively), distributed in numerous, more specified pathways.

Thiamine, as such, appears to be a relevant actor in amino acid synthesis, matching with 10 biomarkers, as well as being strongly involved in phenylalanine, valine, leucine, isoleucine, tyrosine, and lysine metabolisms, more specifically. The overall effect of the vitamin on amino acid metabolism might reflect, overall, its role as a cofactor involved in the Ehrlich pathway, with TPP being essential to the decarboxylation step; however, it might also reflect, indirectly, the relevance of pyridoxine in the transamination of the Ehrlich pathway as well, which can proceed to the conversion of an amino acid towards its associated α-ketoacid, and inversely [42,61]. Since thiamine biosynthesis relies on pyridoxine in case of thiamine deficiencies, in a mechanism that takes priority over all other pyridoxine-dependent reactions [34], it is not impossible that this effect on amino acid synthesis also reflects the indirect influence of pyridoxine. Plus, such an influence of thiamine on both phenylalanine and tyrosine, more specifically, appears consistent with the role held by the vitamin in the pentose phosphate pathway through TPP-dependent transketolases Tkl1/2 [42], from which prephenate is derived, itself a precursor in the biosynthesis of both aromatic amino acids [62]. Similarly, the differences observed regarding both valine and leucine metabolism might reflect the thiamine-dependent character of acetolactate synthase Ilv2 [42], which is involved in the synthesis of both branched-chain amino acids [63]. All in all, such an influence held by thiamine on the amino acid pathways in the *S. cerevisiae* exometabolome appears consistent with our previous observations regarding the wine volatile compounds, which highlighted a similar effect.

The strong influence exerted by high thiamine on carbon metabolism, similarly, affects a large number of pathways involved in the CCM, and highlights, notably, a remarkable effect on keto acids, such as 2-oxocarboxylic acids, C5-branched dibasic acids, and pyruvate, which might reflect the TPP-dependency of the Ehrlich decarboxylation reaction of α-keto acids towards their associated aldehydes [42,61], as well as the conversion of pyruvate towards S-acetolactate through TPP-dependent Ilv2 [42]. The observed influence of thiamine on both the propanoate metabolism and the TCA cycle biomarkers, similarly, might be a reflection of the cofactor role played by TPP in the conversion of α-ketoglutarate to succinyl-CoA by Kgd1 [42], which is further derived towards propanoyl-CoA [64]. This leads to, overall, more evidence of the involvement of thiamine in the CCM, consistent with previous conclusions evaluating its relevance in the volatile compounds that were produced, as well as remarkable compounds, such as acetic acid or glycerol.

In addition, high thiamine appears to affect other notable metabolic pathways, such as those of lipids, as well as having a striking involvement in the synthesis of other cofactors and vitamins, and notably in the metabolism of vitamin B3 through its apparent connection with the nicotinate and nicotinamide metabolism, strongly suggesting the possibility for synergistic effects of vitamins in the yeast metabolism. Interestingly, however, low thiamine seemed to have a similar effect on *S. cerevisiae* metabolic pathways to that of low biotin, likewise impacting riboflavin metabolism in a fashion that still remains unclear.

Thiamine, overall, appeared here as a striking actor in the yeast metabolism, intervening in numerous metabolic pathways, and proving its strong influence on the wine metabolome during alcoholic fermentation. This investigation, on a larger scale, is the first evidence of the impact held by both biotin and thiamine on the wine exometabolome during the alcoholic fermentation by *S. cerevisiae*.

## 4. Conclusions

The effect of both thiamine and biotin on yeasts in an enological context was investigated, from both a kinetic and metabolic standpoint, detailing their influence on the yeast growth and fermentation rates, as well as on the volatile profiles of the final wines. This study has, therefore, provided novel information in regard to the significance of both vitamins in enology, thoroughly assessing the essential character of biotin regarding growth, and the high relevance of thiamine in the course of fermentation processes. Both vitamins have been proven to have a notable influence on the metabolic course of the fermenting yeast, significantly impacting nearly half of the quantified volatile compounds and being further highlighted in an untargeted LC-MS investigation on the wine non-volatile metabolome. A significant role of both thiamine and biotin was found on volatile compounds resulting from the fatty acids and the higher alcohol metabolisms, with the clear role of biotin in the synthesis of fatty acids being highlighted here, and a noticeable influence of thiamine on the central carbon metabolism appearing as well. A striking influence of thiamine on numerous metabolic pathways was also proven here, with its evident role in the wine metabolome being shown here on a large untargeted scale, concomitant with the first evidence of the impact of biotin on the yeast exometabolome. All in all, this exploration of the influence held by both thiamine and biotin on wine composition provides leads for further exploration of the role of vitamins in an enological context, suggesting a possibility for an impact of the wine sensory profiles by their concentrations in the grape musts. As such, varying biotin and thiamine contents of musts may affect wine production, and notably, since thiamine contents of musts were shown to be correlated with their vineyard sites and geographical origin [65], which might factor in inducing differences in the final wines. This, in addition, highlights the evidence of an influence of the yeasts’ physiological state on the management and response to deficiencies, through the appearance of notable differences between the present results, and those resulting from the inoculation of pre-starved yeasts in the medium, as observed in other previous studies. Such an effect, similarly, offers grounds for further understanding of these physiological states when inoculating wine yeasts, evidently a source for relevant oenological applications.

## Figures and Tables

**Figure 1 foods-12-00972-f001:**
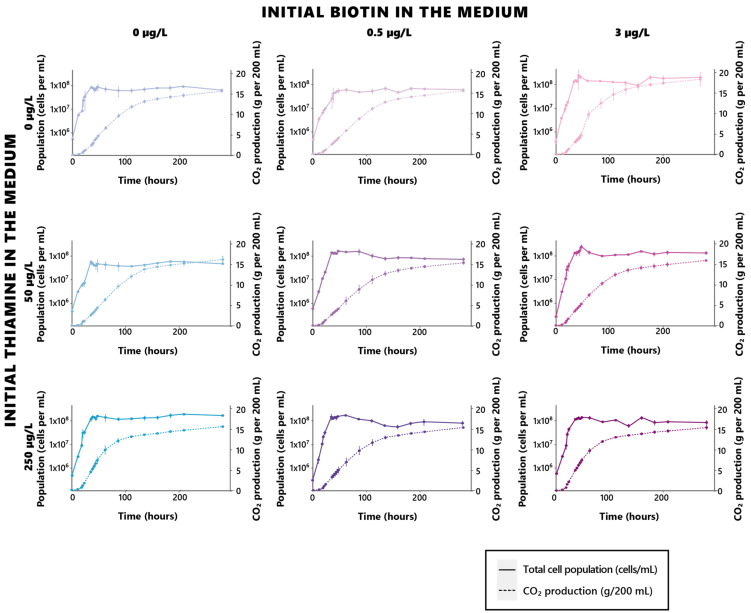
Growth and fermentation profiles of *S. cerevisiae* under different biotin and thiamine concentrations. Bluer and darker blue shades indicate higher initial thiamine in the cultivation medium, while pinker and darker pink shades indicate higher initial biotin in the cultivation medium.

**Figure 2 foods-12-00972-f002:**
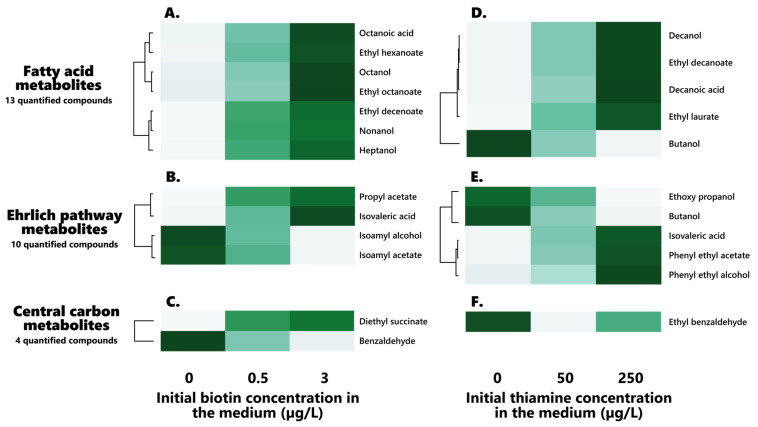
Hierarchical clustering of the biotin (**left**) and thiamine (**right**) significantly impacted volatile compounds in the final wines: effect of both vitamins on their profiles. (**A**–**C**): Biotin-impacted volatile compounds ((**A**): Fatty acid metabolites, (**B**): Ehrlich pathway metabolites, (**C**): Central carbon metabolites). (**D**–**F**): Biotin-impacted volatile compounds ((**D**): Fatty acid metabolites, (**E**): Ehrlich pathway metabolites, (**F**): Central carbon metabolites).

**Figure 3 foods-12-00972-f003:**
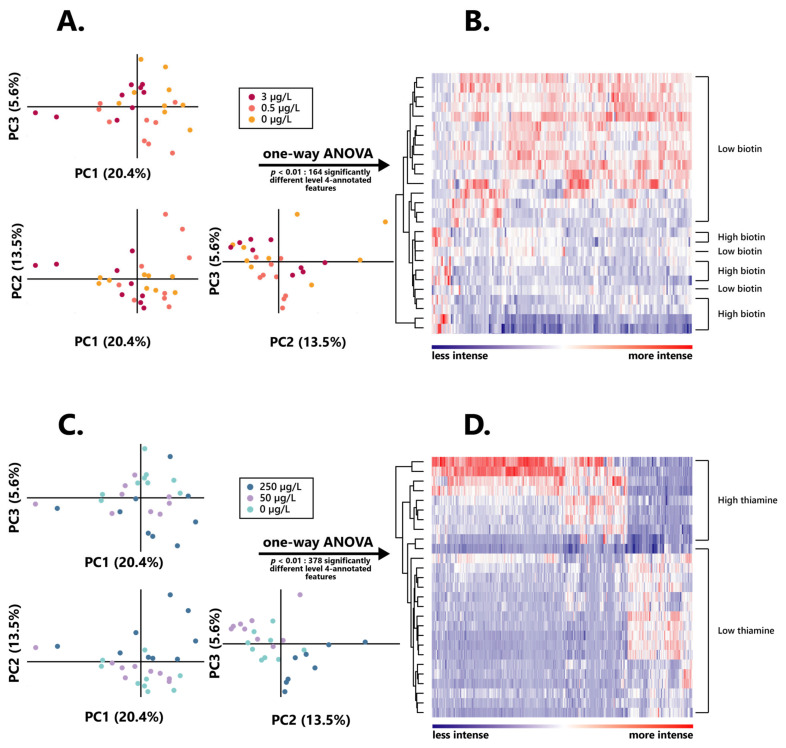
Intensity and distribution of the wine extracted metabolomic features after alcoholic fermentation in varying biotin and thiamine concentrations. (**A**) Principal component analysis individual plot according to the initial biotin content in the growth medium. (**B**) Hierarchical clustered heatmap of the one-way ANOVA significantly different (*p* < 0.01) level 4 annotated features between high and low biotin fermentation conditions. (**C**) Principal component analysis individual plot according to the initial thiamine content in the growth medium. (**D**) Hierarchical clustered heatmap of the one-way ANOVA significantly different (*p* < 0.01) level 4 annotated features between high and low thiamine fermentation conditions.

**Figure 4 foods-12-00972-f004:**
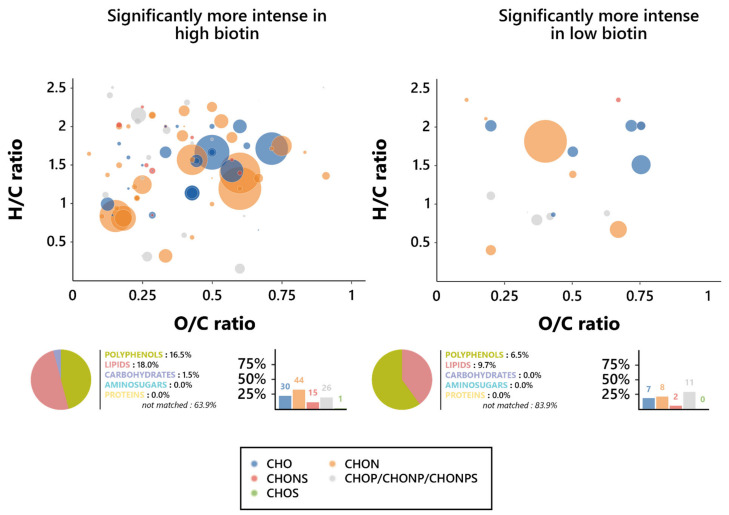
Metabolomic foot-printing of *S. cerevisiae* grown in high or low biotin conditions based on LC-MS-MS untargeted analysis data. Van Krevelen diagrams based on the H/C and O/C atomic ratios for the significantly more intense features (extracted through ANOVA; *p* < 0.01), histogram proportion of their elemental compositions, and pie chart distribution of the predicted families for these biomarkers in high biotin conditions (**left**, 133 markers) and low biotin conditions (**right**, 31 markers). The point size indicates the relative intensity of the extracted masses.

**Figure 5 foods-12-00972-f005:**
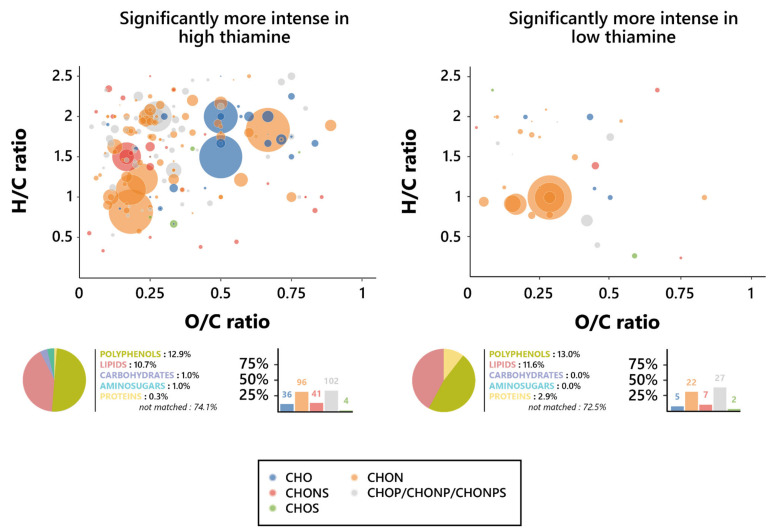
Metabolomic foot-printing of *S. cerevisiae* grown in high or low thiamine conditions based on LC-MS-MS untargeted analysis data. Van Krevelen diagrams based on the H/C and O/C atomic ratios for the significantly more intense features (extracted through ANOVA; *p* < 0.01), histogram proportion of their elemental compositions, and pie chart distribution of the predicted families for these biomarkers in high thiamine conditions (**left**, 378 markers) and low thiamine conditions (**right**, 79 markers). The point size indicates the relative intensity of the extracted masses.

**Table 1 foods-12-00972-t001:** Thiamine and biotin contents of the modified MS300 synthetic musts.

Medium	Thiamine (µg/L)	Biotin (µg/L)
T0	0	0
B1−	50	0
B8−	0	0.5
B1+	250	0
B8+	0	3
B1− B8−	50	0.5
B1+ B8−	250	0.5
B1− B8+	50	3
T+	250	3

**Table 2 foods-12-00972-t002:** Initial biotin impact on the growth and fermentation kinetics.

Parameter	Initial Biotin Concentration (µg/L)		
	0	0.5	3
µ_max_ (h^−1^) ^1^	0.12 ± 0.00 ^a^	0.15 ± 0.01 ^b^	0.16 ± 0.00 ^b^
G (h) ^2^	5.7 ± 0.2 ^a^	4.7 ± 0.2 ^ab^	4.6 ± 0.2 ^b^
y_max_ (cells/mL) ^3^	8.0 × 10^7^ ± 1.7 × 10^7 a^	2.3 × 10^8^ ± 4.6 × 10^7 b^	1.8 × 10^8^ ± 6.5 × 10^7 b^
t_AF_ (h) ^4^	138 ± 11 ^a^	117 ± 11 ^a^	126 ± 11 ^a^
r_CO2max_ (g_CO2_/200 mL/h) ^5^	0.9 ± 0.4 ^a^	1.1 ± 0.8 ^a^	0.8 ± 0.2 ^a^

Note: different letters in the same line indicate significant differences between conditions (Scheirer–Ray–Hare and Dunn tests, *p* < 0.05). ^1^ µ_max_: maximal specific growth rate; ^2^ G: generation time; ^3^ y_max_: maximal population; ^4^ t_AF_: duration of the alcoholic fermentation; ^5^ r_CO2max_: maximal CO_2_ production rate.

**Table 3 foods-12-00972-t003:** Initial thiamine impact on the growth and fermentation kinetics.

Parameter	Initial Thiamine Concentration (µg/L)		
	0	50	250
µ_max_ (h^−1^) ^1^	0.16 ± 0.07 ^a^	0.15 ± 0.06 ^a^	0.13 ± 0.07 ^a^
G (h) ^2^	4.8 ± 0.1 ^a^	4.9 ± 0.1 ^a^	5.3 ± 0.1 ^a^
y_max_ (cells/mL) ^3^	1.5 × 10^8^ ± 6.0 × 10^7 a^	1.4 × 10^8^ ± 8.3 × 10^7 a^	1.1 × 10^8^ ± 3.5 × 10^7 a^
t_AF_ (h) ^4^	136 ± 22 ^a^	134 ± 17 ^a^	112 ± 19 ^b^
r_CO2max_ (g_CO2_/200 mL/h) ^5^	1.1 ± 0.9 ^ab^	0.7 ± 0.1 ^a^	0.9 ± 0.2 ^b^

Note: different letters in the same line indicate significant differences between conditions (Scheirer–Ray–Hare and Dunn tests, *p* < 0.05). ^1^ µ_max_: maximal specific growth rate; ^2^ G: generation time; ^3^ y_max_: maximal population; ^4^ t_AF_: duration of the alcoholic fermentation; ^5^ r_CO2max_: maximal CO_2_ production rate.

**Table 4 foods-12-00972-t004:** Central carbon metabolites in the final wines produced in varying initial biotin concentrations.

Compound	Initial Biotin Concentration (µg/L)		
	0	0.5	3
Acetaldehyde (mg/L)	108.9 ± 32.8 ^a^	144.1 ± 19.8 ^a^	162.4 ± 6.5 ^b^
Acetic acid (g/L)	1.4 ± 0.2 ^a^	1.3 ± 0.4 ^ab^	1.0 ± 0.2 ^b^
D-lactic acid (g/L)	0.2 ± 0.0 ^a^	0.2 ± 0.1^a^	0.2 ± 0.0 ^a^
Glycerol (g/L)	8.0 ± 0.6 ^a^	7.0 ± 0.8 ^b^	6.5 ± 0.3 ^b^
Ethanol (% *v*/*v*)	9.6 ± 0.1^a^	9.6 ± 0.1^a^	9.4 ± 0.3 ^a^

Note: different letters in the same line indicate significant differences between conditions (Scheirer–Ray–Hare and Dunn tests, *p* < 0.05).

**Table 5 foods-12-00972-t005:** Central carbon metabolites in the final wines produced in varying initial thiamine concentrations.

Compound	Initial Thiamine Concentration (µg/L)		
	0	50	250
Acetaldehyde (mg/L)	126.7 ± 37.8 ^a^	135.8 ± 35.8 ^a^	151.7 ± 7.1^a^
Acetic acid (g/L)	1.1 ± 0.4 ^a^	1.2 ± 0.2 ^ab^	1.4 ± 0.2 ^b^
D-lactic acid (g/L)	0.3 ± 0.0 ^a^	0.2 ± 0.0 ^b^	0.2 ± 0.0 ^b^
Glycerol (g/L)	6.8 ± 0.6 ^a^	7.0 ± 0.8 ^a^	7.8 ± 1.06 ^b^
Ethanol (% *v*/*v*)	9.5 ± 0.2 ^a^	9.6 ± 0.1 ^a^	9.5 ± 0.3 ^a^

Note: different letters in the same line indicate significant differences between conditions (Scheirer–Ray–Hare and Dunn tests, *p* < 0.05).

**Table 6 foods-12-00972-t006:** Biotin-dependent volatile contents in the final wines.

Compound (µg/L)	Initial Biotin Concentration (µg/L)		
	0	0.5	3
Benzaldehyde	52.57 ± 16.76 ^a^	36 ± 10.69 ^ab^	29.56 ± 4.93 ^b^
Diethyl succinate	0.34 ± 0.05 ^a^	0.47 ± 0.1 ^ab^	0.44 ± 0.05 ^b^
Ethyl dec-9-enoate	0.69 ± 0.16 ^a^	1.77 ± 3.06 ^ab^	1.19 ± 0.4 ^b^
Ethyl hexanoate	38.29 ± 4.54 ^a^	41.67 ± 7.95 ^a^	52.56 ± 4.33 ^b^
Ethyl octanoate	51.43 ± 4.76 ^a^	54.22 ± 6.7 ^a^	71.56 ± 9.03 ^b^
1-Heptanol	9.14 ± 1.57 ^a^	14.22 ± 4.29 ^b^	14.22 ± 2.44 ^b^
Isoamyl acetate	83 ± 18.15 ^a^	81.44 ± 28.85 ^ab^	59.78 ± 13.46 ^b^
Isoamyl alcohol	75,173.43 ± 8173.63 ^a^	67,535.22 ± 19,964.24 ^a^	49,623.22 ± 4767.42 ^b^
Isovaleric acid	2787.86 ± 156.47 ^a^	3099.56 ± 401.25 ^ab^	3427.22 ± 327 ^b^
1-Nonanol	8.14 ± 1.46 ^a^	11.33 ± 4.06 ^b^	10.44 ± 1.59 ^b^
Octanoic acid	6334 ± 695.2 ^a^	6620.89 ± 1678.83 ^a^	8288.22 ± 730.84 ^b^
1-Octanol	37.71 ± 5.59 ^ab^	35.78 ± 11.02 ^a^	48.56 ± 8.9 ^b^
Propyl acetate	2.26 ± 0.36 ^a^	3.3 ± 0.81^ab^	4.06 ± 0.94 ^b^

Note: different letters in the same line indicate significant differences between conditions (Scheirer–Ray–Hare and Dunn tests, *p* < 0.05).

**Table 7 foods-12-00972-t007:** Thiamine-dependent volatile contents in the final wines.

Compound (µg/L)	Initial Thiamine Concentration (µg/L)		
	0	50	250
1-Butanol	255.5 ± 60.25 ^a^	193 ± 89 ^ab^	152.67 ± 69.36 ^b^
Decanoic acid	1400.25 ± 235.59 ^a^	1463.75 ± 198.13 ^a^	1966.78 ± 343.87 ^b^
1-Decanol	15.59 ± 3 ^a^	17.71 ± 1.72 ^a^	23.29 ± 4.47 ^b^
3-Ethoxy-1-propanol	2002.12 ± 328.39 ^a^	1375.25 ± 200.96 ^b^	1027 ± 263.77 ^b^
Ethyl benzaldehyde	2.89 ± 0.35 ^a^	2.74 ± 0.45 ^a^	3.49 ± 0.33 ^b^
Ethyl decanoate	19.62 ± 4.72 ^a^	23.75 ± 4.3 ^ab^	37.33 ± 11.79 ^b^
Ethyl laurate	2.67 ± 0.93 ^a^	3.36 ± 0.7 ^a^	5.1 ± 1.28 ^b^
Isovaleric acid	2944.5 ± 235.19 ^a^	2973.38 ± 282.2 ^a^	3434.78 ± 455.67 ^b^
Phenyl ethyl acetate	2.67 ± 0.67 ^a^	3.6 ± 0.61 ^a^	8.1 ± 3.15 ^b^
Phenyl ethyl alcohol	20,728.25 ± 4410.14 ^a^	21,323.5 ± 4424.84 ^a^	35,286 ± 9176.91 ^b^

Note: different letters in the same line indicate significant differences between conditions (Scheirer–Ray–Hare and Dunn tests, *p* < 0.05).

## Data Availability

Data are contained within the article and the Appendix A.

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
