# Peer review of "Thiamine and Biotin: Relevance in the Production of Volatile and Non-Volatile Compounds during Saccharomyces cerevisiae Alcoholic Fermentation in Synthetic Grape Must"

_foods, 2023, doi:10.3390/foods12050972_

Round 1

Reviewer 1 Report

Thank you for this complete analysis of the effect of thiamine and biotin on yeasts in wine. It has been very amazing to read about the fermentation rates and the volatile profiles of the wines. You have provided new information about vitamins in enology, which is highly relevant. The key role of biotin in the synthesis of fatty acids has been highlighted here. Contribution about the role of thiamine is also a great point of your work. Please, check the style to avoid us of g.L-1 and g/L. If you use the first, use "x" instead of "."

1. What is the main question addressed by the research? Vitamins are major cofactors to numerous key metabolic pathways in enological yeasts, and both thiamine and biotin, notably, are believed to be essential to yeast fermentation and growth, respectively. In order to further assess and clarify their role in winemaking, and in the resulting wine, alcoholic fermentations of a commercial Saccharomyces cerevisiae active dried yeast were conducted in synthetic media containing various concentrations of both vitamins. Growth and fermentation kinetics were monitored, and proved the essential character of biotin in yeast growth, and of thiamine in fermentation. The synthetic wine volatile compounds were quantified, and notable influences of both vitamins appeared, through a striking positive effect of thiamine on the production of higher alcohols, and of biotin on fatty acids. Beyond the evidence of this influence on fermentations and on the production of volatiles, this work proves, for the first time, the impact held by vitamins on wine yeasts exometabolome, investigated through an untargeted metabolomic analysis. This highlighted chemical differences in the composition of synthetic wines through notably a marked influence of thiamine on 46 named S. cerevisiae metabolic pathways, and especially in amino acids-associated metabolic pathways. This provides, overall, a first evidence of the impact held by both vitamins on the wine.
2. Do you consider the topic original or relevant in the field? Does it
address a specific gap in the field?
Yes, it is relevant.  3. What does it add to the subject area compared with other published
material?
New and specific information about two compounds (Thiamine and biotine) 4. What specific improvements should the authors consider regarding the
methodology? What further controls should be considered?
No additional controls are required   5. Are the conclusions consistent with the evidence and arguments presented
and do they address the main question posed?
Yes, they are consistent   6. Are the references appropriate?
Without any doubt   7. Please include any additional comments on the tables and figures.
Those previously depicted. Uniformize units.

Author Response

Dear Reviewer 1,
Thank you most dearly for your very kind comments, and for your very insightful suggestion regarding the unit style ; as such, we have carefully harmonized the concentrations towards the ‘g/L’ notation.

Reviewer 2 Report

The manuscript by Evers et al. directly addresses the importance of thiamin and biotin in a controlled fermentation setting. This is important in that it can provide insight into postulates that have long been around about the importance of these compounds. The study is limited in that it uses a defined media (not a juice), a sterile environment, and only one yeast, but it still offers important insights. The text is long in certain places and could be reduced. The most interesting thing in my opinion, is that with varying conditions, one did not see stuck fermentations, but rather different chemical and likely sensory outcomes. With that knowledge, it raises the possibility that varying biotin and thiamine concentrations in grapes from specific vineyards could be one factor causing differences in wine. A search of the literature using wine + vineyard + thiamine does identify a recent publication that indicates a correlate between site and thiamine PMID: 33850038). The authors may want to reference this work to further tie their findings to wine making. 

Minor note, it was not clear to me what the a/b notations are in the tables with respect to statistical significance.

Author Response

Dear Reviewer 2,
Thank you most sincerely for all your kind comments. The use of synthetic must media is indeed quite limiting in the way that they do not replicate true winemaking conditions, but, as your mentioned, do offer a first approach of what may happen during wine-like alcoholic fermentations in enology, and were the most accessible mean of controlling both vitamins accessible to the yeast.

Following your suggestion, we have tried to reduce the text a little in some places, and do hope that the changes suit your expectations.

Thank you most dearly for the publication you recommended; we have, as such, referenced the work as suggested in the perspectives of our research, as it does indeed provide insight on how both vitamins may affect wine production in real conditions. We have also found, in a previous study (10.1016/j.foodchem.2022.133860), that vitamins contents of musts might be correlated to their geographical origin, which might indeed take part in the specific signatures of some wines.

The a/b annotations are solely meant to indicate the existence of significant differences (through Dunn tests) between conditions for a given parameter ; as such, the significant effect of the vitamin on the parameter was first verified by Scheirer-Ray-Hare tests, and post-hoc Dunn tests were then performed on each vitamin level for this parameter. The letters do not indicate the level of statistical significance (eg. as the asterisks annotation might), but different letters just mean that the p-value between two levels through Dunn tests is below 0.05.

Reviewer 3 Report

Dear Authors,

 Your research is very interesting the experiments were conducted in a careful and thoughtful manner

I suggest to re-read the article and to do the following minor changes:

1) lines 99 and 126 for Bely and al. and Seguinot and collegues, respectively: please write down the number you have used in the References

2)  lines 143-144 and lines 165 and 166: the temperature of exposure for 30 minutes is 40°C? and for the stirring, the number of rpm?

3)For UHPLC-Q-TOF-MS did you injected the sample as it was or you purified it?

4) Table 3 and table 4 in the line of ymax values  please replace “.” before the 10 with “·”

5) Although the figure 2 is very intuitive, I think it is better to explain in the paper or in the caption of figure 2, the meaning of the various shades of color

6) line 436: ethyl nonanoate, data are not shown

Author Response

Dear Reviewer 3,
Thank you most dearly for your very kind comments! We have made a few adjustments to the manuscript as advised by the other reviewers, and we do hope that these revisions fit your expectations.